# BEX1 is a critical determinant of viral myocarditis

**Colton R. Martens**[1], **Lisa E. Dorn**[1], **Adam D. Kenney**[2,3], **Shyam S. Bansal**[1], **Jacob S. Yount**[2,3], **Federica Accornero**[1,3]*

1 Department of Physiology and Cell Biology, Dorothy M. Davis Heart and Lung Research Institute, The Ohio State University, Columbus, Ohio, United States of America, 2 Department of Microbial Infection and Immunity, The Ohio State University, Columbus, Ohio, United States of America, 3 Infectious Diseases Institute, The Ohio State University, Columbus, Ohio, United States of America

* federica.accornero@osumc.edu

**Data Availability Statement:** Data source file can be found at https://doi.org/10.7910/DVN/NWGDLD.

**Funding:** This work was supported by the National Institute of Health under grants [R01 HL 136951

## Abstract

Viral infection of the heart is a common but underappreciated cause of heart failure. Viruses can cause direct cardiac damage by lysing infected cardiomyocytes. Inflammatory immune responses that limit viral replication can also indirectly cause damage during infection, making regulatory factors that fine-tune these responses particularly important. Identifying and understanding these factors that regulate cardiac immune responses during infection will be essential for developing targeted treatments for virus-associated heart failure. Our laboratory has discovered Brain Expressed X-linked protein 1 (BEX1) as a novel stress-regulated pro-inflammatory factor in the heart. Here we report that BEX1 plays a cardioprotective role in the heart during viral infection. Specifically, we adopted genetic gain- and loss-of-function strategies to modulate BEX1 expression in the heart in the context of coxsackievirus B3 (CVB3)-induced cardiomyopathy and found that BEX1 limits viral replication in cardiomyocytes. Interestingly, despite the greater viral load observed in mice lacking BEX1, inflammatory immune cell recruitment in the mouse heart was profoundly impaired in the absence of BEX1. Overall, the absence of BEX1 accelerated CVB3-driven heart failure and pathologic heart remodeling. This result suggests that limiting inflammatory cell recruitment has detrimental consequences for the heart during viral infections. Conversely, transgenic mice overexpressing BEX1 in cardiomyocytes revealed the efficacy of BEX1 for counteracting viral replication in the heart *in vivo*. We also found that BEX1 retains its antiviral role in isolated cells. Indeed, BEX1 was necessary and sufficient to counteract viral replication in both isolated primary cardiomyocytes and mouse embryonic fibroblasts suggesting a broader applicability of BEX1 as antiviral agent that extended to viruses other than CVB3, including Influenza A and Sendai virus. Mechanistically, BEX1 regulated interferon beta (IFN-β) expression in infected cells. Overall, our study suggests a multifaceted role of BEX1 in the cardiac antiviral immune response.

and R01 HL 154001] and by the US-Israel Binational Science Foundation (BSF) grant number 2017094 to FA. The funders had no role in study design, data collection and analysis, decision to publish, or preparation of the manuscript.

**Competing interests:** The authors have declared that no competing interests exist.

## Author summary

Many viral species infect the heart and cause heart failure, which is an irreversible condition with high prevalence, cost, and morbidity. They damage the heart directly by killing infected cells and indirectly by causing excessive immune reactions. This has made it difficult to develop generalizable treatments, and there is a need to better understand the molecular mechanisms through which the heart naturally defends itself from viruses. Here we have identified the protein Brain Expressed X-Linked 1 (BEX1) as a novel antiviral protein that protects the heart from infection. Mice lacking BEX1 are more susceptible to virus-induced cardiac damage, they have problems recruiting immune cells to the heart, and they have greater difficulty clearing the virus. Conversely, overexpression of BEX1 confers protection from infection. Importantly, BEX1 maintains its antiviral role in isolated cells and in response to a variety of viruses, which indicates that its antiviral effect is broad and not entirely dependent on the immune system. Moreover, it regulates the production of the key antiviral protein interferon beta, which suggests that BEX1 may be an underappreciated master regulator of the antiviral response.

## Introduction

Cardiac viruses are thought to be a common cause of heart failure. Viruses can be detected in about 67 percent of patients with idiopathic dilated cardiomyopathy, and human and animal studies have demonstrated a causal role for many viruses in cardiac dysfunction [1–3]. Among the most prevalent and well-studied cardiotropic viruses in humans is Coxsackievirus B3 (CVB3). By causing cardiomyocyte necrosis and driving inflammation of the myocardium, CVB3 can induce electrical and mechanical cardiac abnormalities overall leading to heart failure [4–6]. Other viruses, including influenza (IAV), parvovirus B19, and cytomegalovirus are known to infect the heart and cause cardiovascular disease [7,8]. Importantly, heart failure is one of the most common co-morbidities in patients infected with the emerging severe acute respiratory syndrome coronavirus 2 (SARS-CoV-2), further emphasizing the importance of understanding the role of viruses in heart failure [9].

Viral myocarditis induced by CVB3 typically progresses in three phases [10]. In the first phase, the virus infects the heart and directly damages the host cells. This includes the lysing of cardiomyocytes [11–13]. In the second phase, the host immune response is activated to combat the virus. Host pattern recognition receptors interact with viral components and upregulate antiviral genes [14–16]. Production of antiviral cytokines recruits immune cells to facilitate pathogen clearance and to initiate tissue repair. This immune phase is necessary to limit viral replication, but immune cells can also cause damage to healthy cardiac tissue in the process [17,18]. The third phase of viral myocarditis is the cardiac remodeling phase. This phase is characterized by fibrotic myocardial scarring and eventual heart failure [19].

A key challenge in viral myocarditis research is understanding how the heart fine-tunes its inflammatory response during infection. Inflammation can contribute to viral clearance, but it is also capable of causing secondary damage to the heart. This has led to both immunostimulants (IFN-β) and immunosuppressants (i.e. prednisone, azathioprine, and cyclosporin) being proposed as treatments [7,20]. Neither have seen widespread utility, and conventional heart failure medication and supportive care remain the dominant treatment [7]. A more detailed understanding of the molecular mechanisms of the cardiac antiviral response is important for the development of more efficient therapeutic strategies.

Our laboratory has previously investigated the role of a poorly studied protein called BEX1 (Brain Expressed X-Linked protein 1) in cardiac inflammation [21]. BEX1 is a member of a small family of intrinsically disordered proteins with largely unknown functions. BEX1 has been implicated in development, neuronal regeneration, cell cycle regulation, and muscle regeneration [22–25]. We have previously seen that BEX1 regulates the inflammatory response in the heart during sterile pressure overload stress [21]. Given the known role of inflammation in both viral clearance and pathogenesis of viral myocarditis, we were prompted to investigate the role of BEX1 in mediating the heart's response to viral infection. Here, using a combination of mouse and cell culture models, we found that BEX1 protects the heart from virus-induced damage and reduces viral abundance. Overall, our work suggests BEX1 as a novel antiviral agent.

## Results

Using a murine model of CVB3-induced viral myocarditis, we found that BEX1 expression was reduced one week after infection (**Fig 1A**), suggesting that BEX1 is regulated by viral stress. To determine if BEX1 is important for the heart's antiviral response, we then infected wild type (WT) and BEX1 knockout (KO) mice with CVB3 for various durations. Despite equal viral load at day two post-infection, viral RNA levels were dramatically higher in BEX1 KO hearts than in WT hearts seven days after infection (**Fig 1B**), suggesting that BEX1 is necessary to limit viral proliferation. CVB3 was undetectable in both genotypes 28 days post-infection, suggesting the virus does not persist long term in the myocardium (**Fig 1B**). In addition, assessment of myocardial viral presence by immunostaining revealed that BEX1 KO hearts had a significantly higher percentage of cardiomyocytes infected with CVB3 (**Fig 1C and 1D**). Taken together, these results show that BEX1 serves an antiviral role and restricts CVB3 replication in mouse hearts.

Our previous work on the role of BEX1 in the context of sterile cardiac injuries hinted at BEX1 as an immune modulator [21]. We therefore investigated whether BEX1 regulates virus-induced myocardial infiltration of immune cells. We first stained cardiac sections over the course of infection with hematoxylin and eosin (H&E) to track histopathological changes (**Fig 2A**). Cell infiltration into the myocardium was evident in both genotypes by day seven post-infection, as demonstrated by the abundance of mononuclear cells (**Fig 2A**). Given the role of infiltrating immune cells in limiting viral proliferation, we then measured different immune cell populations in infected WT and BEX1 KO hearts using flow cytometry. The abundance of total leukocytes (CD45+) was significantly lower at seven days post-infection in the BEX1 KO hearts (**Fig 2B and 2C**), suggesting that BEX1 KO mice have impaired immune cell recruitment. Similar significant results were seen at day 7 for T-cells and total monocytes (**Fig 2D and 2E**), while other immune cell types showed only a trend towards reduction in BEX1 KO hearts or no change (**Fig 2F–2K**). Together, these observations suggest that BEX1 plays an important role in myocardial immune cell recruitment and regulation during CVB3 infection. However, analysis of pro-inflammatory gene program activation did not show a unidirectional regulation by BEX1. While induction of cytokines such as IFN-γ and CCL5 was impaired or delayed in the infected BEX1-deficient hearts (**S1A–S1C Fig**), the expression of others such as CXCL1 and CXCL13 was greater (**S1K–S1L Fig**). The latter could suggest a direct attempt of hearts lacking BEX1 to compensate for the observed uncontrolled viral replication by up-regulating inflammatory cell recruitment. Overall, our results also indicate that expression of pro-inflammatory genes in BEX1 KO mice is insufficient to compensate for the defective immune cell recruitment in the heart.

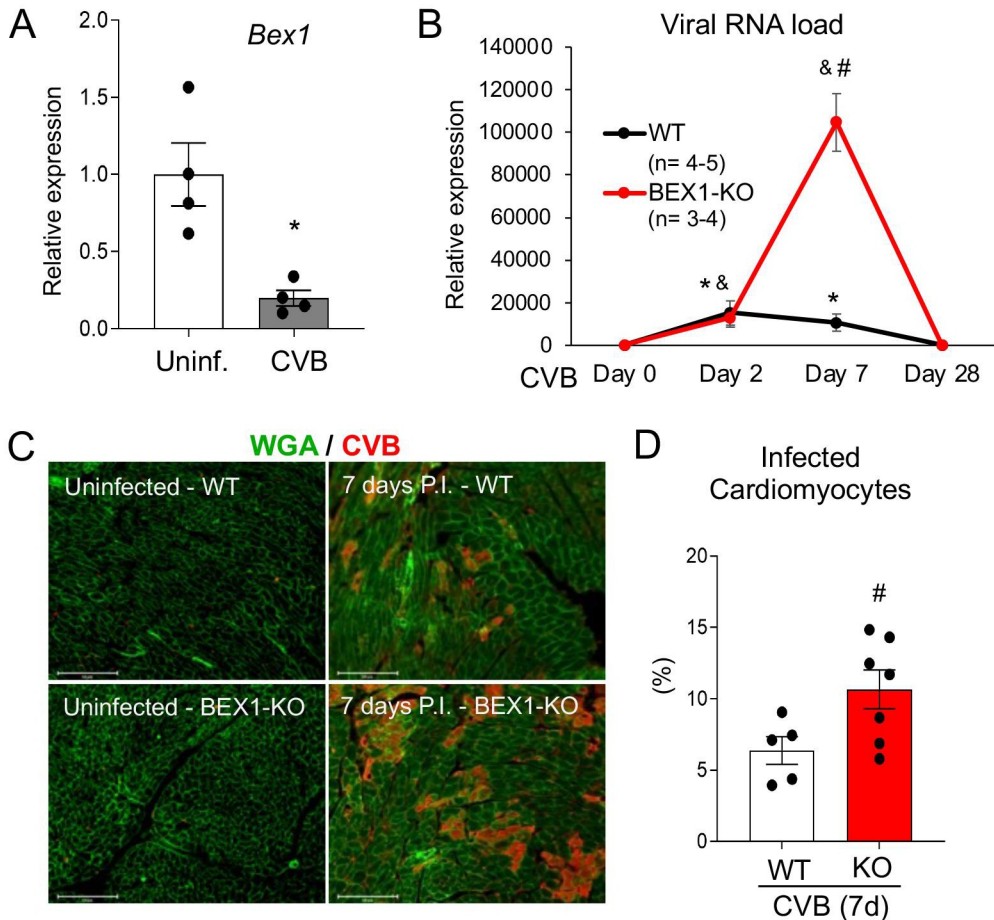

**Fig 1. BEX1 limits viral infection *in vivo*.** A) qPCR for BEX1 mRNA expression (normalized to Rpl7 mRNA) in WT mouse hearts in uninfected conditions and one week following infection with CVB3. B) qPCR for the CVB3 RNA genome (relative to Rpl7) in WT and BEX1 KO hearts that were not infected or infected for 2, 7, and 28 days with CVB3. C) Immunostaining of WT and BEX1 KO cardiac sections in uninfected conditions and one week following infection with CVB3. Replicating CVB3 was detected using a double-stranded RNA-binding antibody and is shown in red, while the cell outlines were stained in green with wheat germ agglutinin stain. Scale bar = 125μm. D) Quantification of the percent of cardiomyocytes that were infected in WT and BEX1 KO cardiac sections shown in panel C. Statistical analyses: t-test for panels A and D; 2-way ANOVA for panel B. (* $p < 0.05$ CVB3-treated WT vs. uninfected WT; # $p < 0.05$ BEX1 KO vs. WT same treatment).

We next sought to determine the physiological consequences of the loss of BEX1 during infection. Body weight decreased over the course of infection but did not differ significantly between genotypes (**Fig 3A**). Similarly, normalized heart and spleen weight did not differ between genotypes over the course of infection (**Fig 3B and 3C**). However, cardiac ejection fraction was significantly lower in BEX1 KO mice than in WT mice at four weeks post-infection (**Fig 3D** and **S1 Table**), suggesting the BEX1 KO hearts were more susceptible to virus-induced cardiac dysfunction. As fibrosis is a hallmark of pathological cardiac remodeling post-infections, we utilized Masson's trichrome staining to measure fibrosis and found that BEX1 KO hearts developed significantly more fibrotic tissue area by the fourth week of infection (**Fig 3E and 3F**). These results suggest that BEX1 is cardioprotective during viral infection.

To test if BEX1 overexpression would protect the heart against viral infection, we then adopted an *in vivo* model of cardiomyocyte-specific BEX1 overexpression (**Fig 4A**). One week after CVB3 infection, the BEX1 overexpressing mice had significantly reduced viral load as

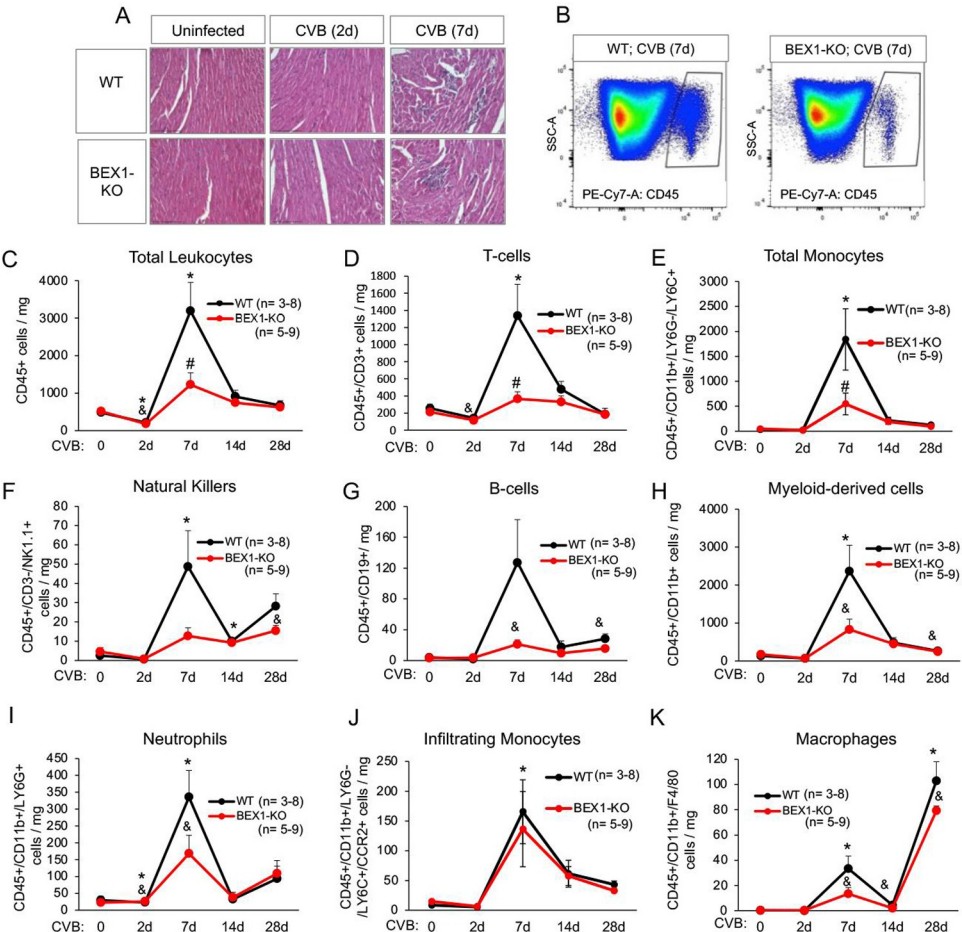

**Fig 2. Immune cell infiltration is impaired in BEX1 KO hearts during CVB3 infection.** A) Hematoxylin and Eosin staining was conducted in WT and BEX1 KO mice following the indicated treatments. Scale bar = 125μm. B) The gating of CD45+ cells in representative WT and BEX1 KO 7 days post-infection hearts is shown for total leukocyte abundance using flow cytometry. D-K) Total Leukocytes (C), T lymphocytes (D), total monocytes (E), natural killers (F), B lymphocytes (G), myeloid-derived cells (H), neutrophils (I), infiltrating monocytes (J), and macrophages (K) were evaluated in WT and BEX1 KO hearts at the indicated post-infection time points using flow cytometry. d = days. Statistical analyses by 2-way ANOVA. (* $p < 0.05$ CVB3-treated WT vs. uninfected WT; & $p < 0.05$ CVB3-treated BEX1 KO vs. uninfected BEX1 KO; # $p < 0.05$ BEX1 KO vs. WT same treatment).

compared to their WT counterparts (**Fig 4B**). These results further confirm that BEX1 is a key antiviral factor for the heart. No cardiac dysfunction was observed in either control or BEX1 transgenic mice at 28 days post-infection (**S2A–S2D Fig** and **S2 Table**). Interestingly, no genotype-dependent differences in immune cell recruitment were observed with BEX1 overexpression in cardiomyocytes (**S2E-S2J Fig**). This result suggests that BEX1 could exert antiviral effects independent from the immune system. To test if indeed BEX1 plays a cell autonomous role in limiting viral replication in cardiomyocytes, we then overexpressed BEX1 in isolated primary rat neonatal cardiomyocytes (**Fig 4C**) and infected them with CVB3. In comparison to β-galactosidase-overexpressing controls, BEX1 overexpressing cardiomyocytes had reduced viral abundance 12 hours after infection (**Fig 4B**). These results further confirm that BEX1 is a key antiviral factor for cardiomyocytes.

In order to determine if BEX1 could protect cells other than cardiac from viral infection, and whether or not BEX1 could protect against viruses other than CVB3, we isolated WT and

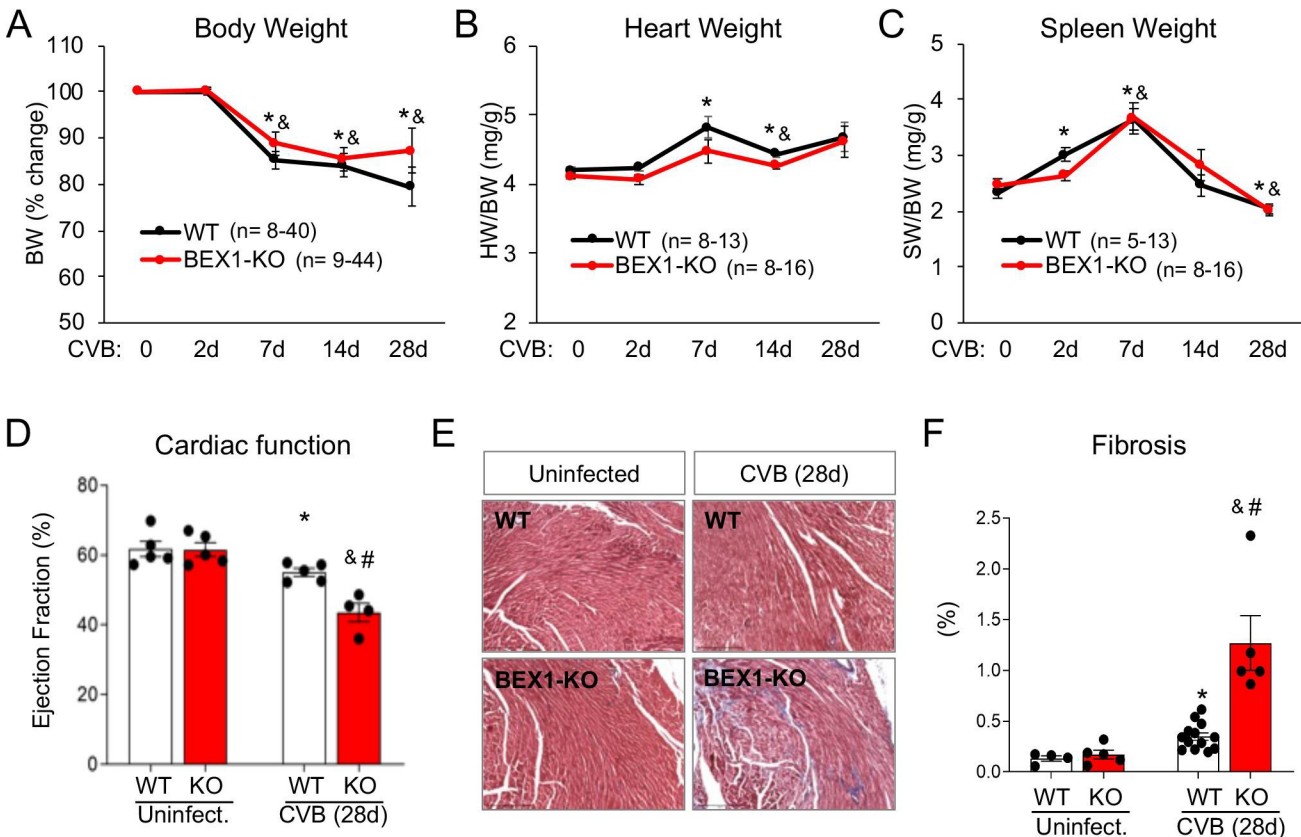

**Fig 3. Loss of BEX1 increases CVB3-induced cardiac damage.** A) Percent change in body weight of WT and BEX1 KO mice over the course of infection is shown. B-C) Heart weight (B) and spleen weight (C) (both normalized to body weight) of each genotype over the course of infection is shown. D) Percent ejection fraction was measured in WT and BEX1 KO mice in uninfected conditions and four weeks following infection. E-F) Cardiac fibrosis was evaluated using Masson's trichrome stain (scale bar = 200μm) (E), and it was quantified using ImageJ (F). Statistical analyses by 2-way ANOVA. (* $p < 0.05$ CVB3-treated WT vs. uninfected WT; & $p < 0.05$ CVB3-treated BEX1 KO vs. uninfected BEX1 KO; # BEX1 KO vs. WT same treatment).

BEX1 KO mouse embryonic fibroblasts (MEFs) and infected them with three different RNA viruses for 24 hours. We found that BEX1 KO MEFs are significantly more susceptible to infection by CVB3, as well as by influenza A virus (IAV) and Sendai virus ([SeV]; a murine parainfluenza virus) (**Fig 5A–5C**). Previous work from our group demonstrated a role for BEX1 in regulating AU-rich element containing transcripts [21]. Interferon genes belong to this category and are critical mediators of viral responses. We found defective interferon beta (IFN-β) expression in infected BEX1-null cells (**Fig 5D–5F**), and a rescue of their increased viral load following IFN-β supplementation (**Fig 5G–5I**). These results suggest that BEX1 can serve a direct role in limiting viral infection by regulating IFN-β. Altogether we discovered BEX1 as a novel determinant of cardiac viral infection that limits viral replication, and that it is a broad antiviral regulator beyond CVB3 and the heart.

## Methods

### Ethic statement

This work was approved by the Institutional Animal Care and Use Committee at The Ohio State University.

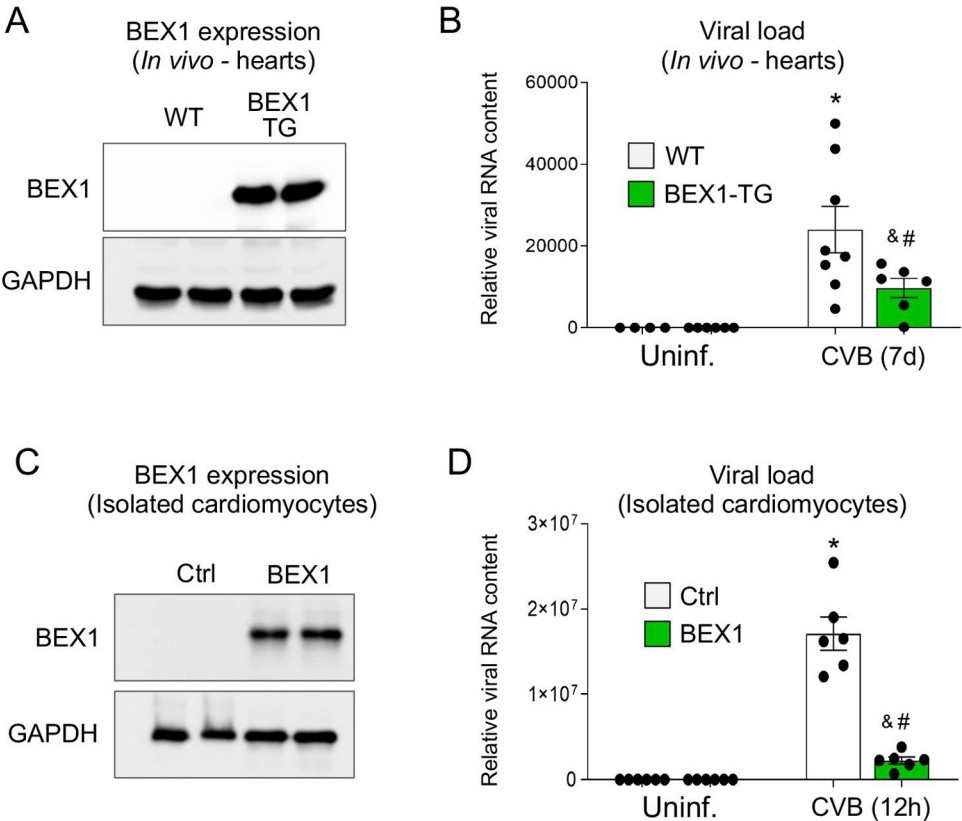

**Fig 4. BEX1 overexpression protects against CVB3 infection:** A) Western blot for BEX1 and GAPDH (loading control) in isolated neonatal rat cardiomyocytes. B) qPCR showing abundance of CVB3 RNA (normalized to Rpl7) 12 hours after infection (MOI = 1). C) Western blot for BEX1 and GAPDH (loading control) in cardiac protein extracts from WT and BEX1 transgenic mice. D) qPCR showing abundance of CVB3 RNA (normalized to Rpl7) in hearts of uninfected or infected (seven days) WT and BEX1-overexpressing mice. Statistical analyses by 2-way ANOVA. (* $p < 0.05$ CVB3-treated WT vs. uninfected WT; & $p < 0.05$ CVB3-treated BEX1 KO vs. uninfected BEX1 KO; # $p < 0.05$ BEX1 KO vs. WT same treatment).

## Virus propagation, Infection, and quantification

Coxsackievirus B3 (CVB3) was kindly provided by Jeffrey Bergelson from the University of Pennsylvania School of Medicine and propagated in HeLa cells at 37°C, 5% $CO_2$ for 72 hours, concentrated by ultracentrifugation through a 20% sucrose cushion, and titered on HeLa cells. Influenza A virus A/PR/8/34 (H1N1) expressing green fluorescent protein (PR8-GFP) was propagated in 10-day-old embryonated chicken eggs (Charles River) for 48 hours at 37°C and titered on MDCK cells. Sendai virus expressing green fluorescent protein (SEV-GFP) was propagated in 10-day-old embryonated chicken eggs at 37°C for 40 hours and titered on Vero cells. Infections were carried out for 24 hours with PR8-GFP or SEV-GFP at an MOI of 1.0. Infection percentages were calculated by detecting virus-encoded GFP via flow cytometry on a FACSCanto II flow cytometer (BD Biosciences). Data were analyzed using FlowJo software.

## Mice

6–10 week old wild type and BEX1 KO mice of 129Sv/Ev background and littermate WT controls were used. The BEX1 KO was global and was generated as described previously [25].

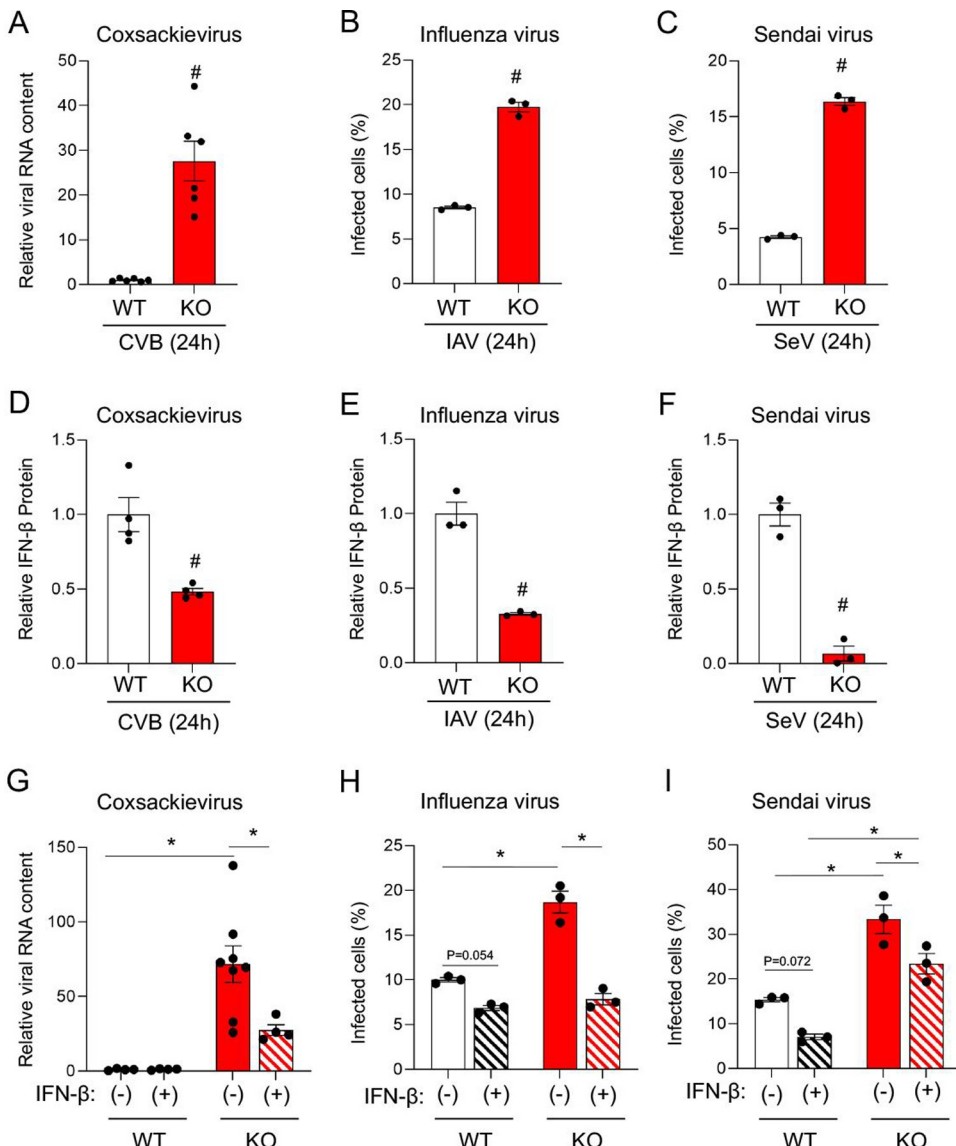

**Fig 5. Loss of BEX1 increases susceptibility to multiple viruses *ex vivo*.** A) qPCR of the CVB3 genome (normalized to Rpl7) from WT and BEX1 KO mouse embryonic fibroblasts (MEFs) infected with CVB3 (MOI = 1) for 24 hours. B-C) Percent of MEFs (WT or BEX1 KO) infected with influenza (B) or Sendai virus (C) after 24 hours, as measured by flow cytometry of GFP-tagged viral capsids. D-F) ELISA assessing IFN-β protein present in growth medium of WT or BEX1 KO MEFs 24 hours after infection with CVB3 (D), influenza (E), or Sendai virus (F). G-I) Rescue of antiviral function in BEX1 KO MEFs by treatment with recombinant IFN-β for cells infected with CVB3 (G), influenza (H), or Sendai virus (I). Statistical analyses by t-test (A-F) and two-way ANOVA (G-I). (# $p < 0.05$ BEX1 KO vs. WT same treatment; * $p < 0.05$ between indicated groups).

BEX1-overexpressing mice were obtained using α-myosin heavy chain as the driving promoter as previously described [21], and used in the C57BL6/J background alongside with their littermate WT controls at 6–8 weeks of age. Mice were infected by intraperitoneal injection with $10^3$ infectious units of CVB3. Mice were infected between 6 and 8 weeks of age (the age at which the CVB3 receptor is highly expressed in the heart). Mice were sacrificed by removal of the heart while sedated with isoflurane.

## Flow cytometry

Cardiac mononuclear cells were isolated as described previously [26]. Briefly, hearts were harvested, washed with PBS, and accurately weighed. Using a heavy-duty single-edged blade, hearts were finely minced in a petri-dish and transferred to 50 mL tubes followed by the addition of 3 ml of cold PBS. Tubes were centrifuged at 100 xg for 2 minutes, and the supernatant was carefully decanted. Collagenase II solution in DMEM (1 mg/mL; 7–8 mL) was added to each tube followed by incubation at 37˚C for 20 min for tissue-digestion. Tubes were mixed every 5–6 min to re-suspend tissues. The digestant was filtered through 40 μm filters in fresh 50 mL tubes containing cold PBS supplemented with 0.5% (w/v) BSA and 1 mM EDTA. Undigested tissue-chunks were lightly triturated using plunger of a 3 mL syringe to ensure complete digestion. The filtrate was centrifuged at 500 xg at 4˚C for 10 minutes and the supernatant was carefully decanted. Cell-pellets were re-suspended in ~200 μL cold staining buffer (PBS supplemented with 0.5% w/v BSA and 1 mM EDTA) and fixed using 200 μL of 1% PFA. Tubes were incubated on ice for 10 min to ensure complete fixing and the excess PFA was neutralized using 1.8 mL of cold staining buffer. Fixed cells were pelleted at 4˚C by centrifugation at 500 xg for 10 min. The supernatant was decanted, and cells were re-suspended in ~200–300 μL staining buffer. Fixed samples were stored at 4˚C until used for staining.

A cocktail of all the extracellular antibodies was prepared and added to all the samples followed by incubation on ice for one hour. Antibodies used were: Invitrogen 25045182 (CD45 rat IgG2b kappa, PE-cyanine7), Invitrogen 63011280 (CD11b rat IgG2b kappa, Super Bright 600), Invitrogen 17003280 (CD3 rat IgG2b kappa, APC), Invitrogen 45480180 (F4/80 rat IgG2a kappa, PerCP cyanine5.5), BioLegend 108707 (NK1.1 mouse IgG2a kappa, PE), BioLegend 127651 (LY6G rat IgG2a kappa, APC/Fire), and Invitrogen 48019382 (CD19 rat IgG2a kappa, eFluor 450). Isotype control antibodies used were: Invitrogen 25403181 (rat IgG2b kappa, PE-cyanine7), Invitrogen 63403180 (rat IgG2b kappa, Super Bright 600), Invitrogen 17403181 (rat IgG2b kappa, APC), Invitrogen 45432180 (rat IgG2a kappa, PerCP cyanine5.5), BioLegend 400211 (mouse IgG2a kappa, PE), BioLegend 400567 (rat IgG2a kappa, APC/Fire), and Invitrogen 48432182 (rat IgG2a kappa, eFluor 450). To wash excess antibodies, 4 mL of cold PBS was added, and cells were pelleted by centrifugation at 500 xg for 10 min. The supernatant was decanted, and the cell pellets were reconstituted in 100–200 μL of cold PBS. For accurate cell counting, 2 μl Accucount beads (Spherotech ACBP-100-10) were added to each sample. Data was acquired using a BD Fortessa flow cytometer and was analyzed using FlowJo software. An initial gate was drawn from all detected events, and single cells were isolated from that gate to remove doublets. CD45 positive leukocytes were separated from CD45 negative cells and were further gated into CD19 positive B-cells, CD11b positive myeloid-derived cells, and CD3 positive T-cells. CD3 negative cells were further gated into Nk1.1 positive natural killer cells. CD11b positive cells were further gated into LY6G positive neutrophils and F4/80 positive macrophages. Isotype controls were used in each case to determine which cells were positive or negative for a given marker. Gated events were normalized to the internal control beads and to the initial mass of the heart to derive normalized cells per milligram of tissue, as described previously [26].

## RNA expression analysis

Total RNA was isolated from cells or tissue using TRIzol Reagent. RNA was converted to cDNA using the High Capacity cDNA Reverse Transcription kit (Applied Biosystems), and quantitative PCR was performed using SYBR green reagent (Applied Biosystems). Transcript abundance was normalized to abundance of RPL7 mRNA. In experiments with multiple time points and multiple genotypes, expression was displayed relative to the first wild type time

point. Primers used were: *mBex1* 5'-AGGAGAAGGCAAGGATAGGC-3', 5'-TTCTGATGG-TATCTTGTGGCTTT-3'

 *mRpl7* 5'-TGGAACCATGGAGGCTGT-3', 5'-CACAGCGGGAACCTTTTTC-3'

 *rRpl7* 5'-AAAAGAAGGTTGCCGCTG-3', 5'-TAGAAGTTGCCAGCTTTCC-3'

 *CVB3 Vp1* 5'-GGGTCACACGTCACAAGTAGTG-3', 5'-GTCAGCTCCAGGTCGAACC-3'

## Western blot and cytokine array

Total protein was isolated from cells or tissue using RIPA buffer (150 mM NaCl, 150 mM TRIS, 1% IGEPAL, 0.5% DOC, 0.1% SDS, pH = 8.0). Isolated protein was placed in SDS loading buffer and was boiled at 90˚C for 10 minutes prior to loading. Samples were run on 10% polyacrylamide gels and transferred onto PVDF membranes (BioRad 1620177) using the semi-dry transfer method. Membranes were blocked in 5% milk in TBS-T for one hour and placed in primary antibody at 4˚C overnight. Membranes were washed and placed in HRP-conjugated secondary antibody for two hours at room temperature. Membranes were washed again and covered with a 1:1 mix of Supersignal West Fempto Stable Peroxide Buffer and Supersignal West Fempto Luminol/Enhancer Solution (Thermo Scientific 1856190 and 1856189). Antibodies used were against BEX1 (custom made by Dr. Frank Margolis, University of Maryland; rabbit polyclonal; diluted 1:10,000) and GAPDH (Fitzgerald Industries; R-G109a; mouse monoclonal; diluted 1:20,000). The Cytokine Array (R&D Systems; ARY006) was performed following manufacturer instruction by incubating a pool of four hearts per condition and genotype on each membrane.

## Cell culture

Wild type and BEX1 KO mouse embryonic fibroblasts (MEFs) were isolated from day 15.5 to 16.5 embryos as previously described [21]. The MEFs were grown in Dulbecco's Modified Eagle's Medium in 10% bovine growth serum and 1% penicillin-streptomycin mix in 5% $CO_2$. Rat neonatal cardiomyocytes were obtained from rat pups as described previously [27]. In brief, hearts were harvested from two-day-old Sprague-Dawley outbred rat pups, digested overnight in trypsin, and dissociated further in collagenase the next day. The digestion was filtered, and sedimented, and cardiomyocytes were plated in M199 medium in 15% bovine growth serum and 1% penicillin-streptomycin in 5% $CO_2$ overnight. Thereafter, cardiomyocytes were cultured in M199 medium without serum, but with 1% penicillin-streptomycin in 5% $CO_2$ for 24 hours prior to 12 hours of CVB3 infection. BEX1 and β-galactosidase were overexpressed by transducing cardiomyocytes with adenoviral vectors. CVB3 infections were always carried out with a multiplicity of infection of 1. For the IFN-β rescue experiments, cells were exposed to human recombinant IFN-β (EMD Millipore IF014) in growth media at 40 U/ml for 24 hours prior to infection. IFN-β-containing media was left on the cells during the subsequent infection.

## Echocardiography

Mice were lightly anesthetized using 1.5% isoflurane, and echocardiographic measurements were taken using a Vevo2100 Visual Sonics (Visual Sonics) system and MS-400 transducer. Measurements were taken in the M-mode using the parasternal short-axis view at the level of the papillary muscles. Percent ejection fraction was determined from averaging at least three consecutive cardiac cycles using the VevoLabs software.

## Histology and immunostaining

Hematoxylin and eosin, and Masson's trichrome staining for fibrosis were performed from heart histological sections generated with paraffin embedding. For immunostaining the hearts

were placed in 1% paraformaldehyde (in PBS) for 1 hour at room temperature. They were then placed in 30% sucrose in PBS at 4°C overnight. It was then embedded and frozen in Optimal Cutting Temperature compound (OCT), cut into 7 μm sections, and mounted on slides. The slides were stored at -80°C until ready for staining. Prior to staining, slides were warmed at room temperature for 10 minutes. They were then washed in PBS for 15 minutes and were blocked (5% BGS, 0.1% Triton, 0.01% sodium azide in PBS) for 1 hour. They were then incubated with the primary antibody (anti-dsRNA antibody; Kerafast ES2001) at 4°C overnight. The next day, the slides were washed in PBS and incubated for two hours in secondary antibody and wheat germ agglutinin stain to visualize the cell outline. Slides were then washed again in PBS, dried briefly, and set with VectaShield Hard Set.

## Statistical analysis

Results are presented as mean ± SEM. Two-tailed student's t-test was used for comparisons between two groups, and two-way ANOVAs with subsequent Tukey post hoc tests were used for comparisons involving three or more groups. A p-value of 0.05 or lower was considered significant.

## Discussion

Viral myocarditis can be a serious condition that results in sudden cardiac death or onset of heart failure [6,28]. There is currently no cure, and the development of efficient treatments requires a more detailed understanding of the cardiac antiviral response and the factors involved. In this study, we report that BEX1 is an antiviral regulator in the heart *in vivo*. This is the first study to find an antiviral role for BEX1. Importantly, we saw that BEX1 restricts infection with multiple different viruses, which could suggest that BEX1 is a broad mediator of intrinsic cellular antiviral responses. The three viruses studied here (IAV, SeV, and CVB3) are all single-stranded RNA viruses, and it will be interesting to determine whether BEX1 affects DNA viruses and other RNA virus families. Importantly, SARS-CoV-2 is also a single-stranded RNA virus, and testing in the future whether BEX1 limits coronavirus infection is of extreme importance.

Our results also show that BEX1 serves an antiviral role in both cardiomyocytes and mouse embryonic fibroblasts *ex vivo*. This could suggest that BEX1 serves an antiviral role in multiple tissues, and the range of BEX1's antiviral activity should be evaluated in future studies. Importantly, BEX1 was shown to regulate immune cell recruitment *in vivo*, while still retaining antiviral activity in isolated cells. This indicates that BEX1 regulates the antiviral response at multiple points, which is consistent with a broad regulatory mechanism. We have previously shown that BEX1 can regulate the stability of pro-inflammatory mRNAs that contain AU-rich elements (AREs) in their 3' untranslated region [21]. AREs usually destabilize the transcript under normal conditions to limit protein expression [29]. Under conditions of stress, the ARE-binding proteins are altered such that transcript stability is increased [30,31]. This allows rapid induction of proteins during stress. Many pro-inflammatory transcripts contain AREs, and the fact that BEX1 regulates the antiviral response at multiple points could be explained by the stabilization of multiple antiviral ARE-containing mRNAs [32–35]. Indeed, interferon genes contain AREs and we observed defective induction of IFN-γ in BEX1-deficient hearts. Additionally, we found that BEX1 KO cells had impaired production of IFN-β during infection, and addition of recombinant IFN-β could partially rescue the antiviral defect. A reason for the fact that rescue of BEX1-null cells by IFN-β was only partial in the context of CVB3 and Sendai infections can be speculated upon based on a recent study on another member of the BEX family of proteins, BEX3 [36]. This study discovered the ability of BEX3 to bind double

stranded RNA, raising the intriguing possibility that BEX1 could share the same property and directly interfere with viral RNA replication [36]. It is also possible that the BEX1-dependent modulation of interferon can only exert full effects in an *in vivo* context, where interplays between the infected host and the immune system are occurring.

Both the current study and our previous work find that BEX1 regulates immune cell recruitment and inflammation in the heart. In addition to its importance during cardiac viral infection, inflammation also contributes to many sterile cardiomyopathies, including arrhythmias, myocardial infarction, metabolic cardiomyopathy, and aging-associated cardiac dysfunction [37–40]. The progression of these diseases often depends on infiltrating immune cells, and the fact that BEX1 regulates the abundance of myocardial immune cells during injury could suggest that BEX1 is a common regulator of many cardiomyopathies [41]. This is supported by our previous study, where BEX1 was shown to drive cardiac dysfunction resulting from pressure overload stress. However, the finding that BEX1 can limit viral replication independent from immune cell recruitment hints at a dominant role for this protein in protecting the infected host in a cell autonomous way. To note, a current limitation of this study is the use of a global loss of function mouse model, where we cannot exclude a contribution of BEX1 in determining the behavior of immune cells. With this in mind, it is still possible that BEX1 is important in both cardiomyocytes and non-myocyte cell populations in the context of viral infections *in vivo*. Future studies should characterize the cell specificity role of BEX1 in complex physiological systems such as the heart.

The development of therapeutics for viral myocarditis has been hindered by the fact that different stages of the disease may require different treatment strategies. For example, immunostimulants may be beneficial during early stages when most of the damage is caused directly by the virus, but they may be harmful in later stages where damage is primarily caused by excessive inflammation. Conversely, anti-inflammatory medications may be harmful in early stages but beneficial in later stages. Not surprisingly, broad therapeutic approaches aimed at either enhancing or reducing inflammation have often been unsuccessful, and it is clear that they fail to account for important nuances [7]. Factors such as the stage of viral myocarditis, the particular cardiac virus involved, and unique contextual factors impacting the biology of the affected individuals can all influence what treatment strategy is best. In this regards it is possible that BEX1 operates at multiple time points of infection, where its role on the immune system and the one directly affecting viral replication in the host could be uniquely important at different infectious disease stages. While future work will be needed to further our understanding on the mechanistic insights of BEX1 action in different cell types, our results identify BEX1 as a previously unrecognized player involved in the response to viral infections.

## Supporting information

**S1 Fig. Cytokine expression in WT and BEX1 KO hearts during CVB3 infection.** A) Cytokine array representative blots where each membrane was incubated with four biological heart protein extract replicates from the indicated genotypes and treatments. B-V) Quantification of the cytokines observed in the heart of WT and BEX1 KO throughout the time course of CVB infection.
(PDF)

**S2 Fig.** Effects of BEX1 overexpression on CVB-induced cardiac pathology: A-C) WT and BEX1-TG mice were compared in terms of their relative heart weight (A), percent ejection fraction (B), and cardiac fibrosis (C) in uninfected conditions and following 28 days of CVB infection. E-J) Flow cytometry showing the abundance of CD45+ total leukocytes (E), CD11b + myeloid-derived cells (F), CD11b+/F4/80+ macrophages (G), CD11b+/LY6G+ neutrophils

(H), CD11b-/CD3+ T lymphocytes (I), and CD19+ B lymphocytes (J) in the hearts of WT and BEX1-TG mice in uninfected conditions and following seven days of infection. (Error bars represent SEM; * p<0.05 CVB-treated WT vs. uninfected WT; & p<0.05 CVB-treated BEX1-TG vs. uninfected BEX1-TG; # p<0.05 BEX1-TG vs. WT same treatment).
(PDF)

**S1 Table. Echocardiographic measurements in uninfected and infected WT and BEX1-KO mice (mean ± standard error).** IVS;d = interventricular septum thickness; diastole. IVS; s = interventricular septum thickness; systole. LVID;d = left ventricular internal diameter; diastole. LVID;s = left ventricular internal diameter; systole. LVPW;d = left ventricular posterior wall thickness; diastole. LVPW;s = left ventricular posterior wall thickness; systole. EF = ejection fraction. FS = fractional shortening. LV Vol;d = left ventricular volume; diastole. LV Vol;s = left ventricular volume; systole. SV = stroke volume. HR = heart rate. (# = p<0.05 WT Day 28 vs. BEX1-KO Day 28; * = p<0.05 WT Day 28 vs. WT Uninfected; & = p<0.05 BEX1-KO Day 28 vs. BEX1-KO Uninfected.
(PDF)

**S2 Table. Echocardiographic measurements in uninfected and infected WT and BEX1-TG mice (mean ± standard error).** IVS;d = interventricular septum thickness; diastole. IVS; s = interventricular septum thickness; systole. LVID;d = left ventricular internal diameter; diastole. LVID;s = left ventricular internal diameter; systole. LVPW;d = left ventricular posterior wall thickness; diastole. LVPW;s = left ventricular posterior wall thickness; systole. EF = ejection fraction. FS = fractional shortening. LV Vol;d = left ventricular volume; diastole. LV Vol;s = left ventricular volume; systole. SV = stroke volume. HR = heart rate. (# = p<0.05 WT Day 28 vs. BEX1-TG Day 28; * = p<0.05 WT Day 28 vs. WT Uninfected; & = p<0.05 BEX1-TG Day 28 vs. BEX1-TG Uninfected.
(PDF)

## Author Contributions

**Conceptualization:** Shyam S. Bansal, Jacob S. Yount, Federica Accornero.

**Data curation:** Colton R. Martens, Lisa E. Dorn, Adam D. Kenney.

**Formal analysis:** Colton R. Martens, Lisa E. Dorn, Adam D. Kenney.

**Funding acquisition:** Federica Accornero.

**Investigation:** Colton R. Martens, Adam D. Kenney.

**Methodology:** Colton R. Martens, Shyam S. Bansal, Jacob S. Yount.

**Project administration:** Federica Accornero.

**Supervision:** Federica Accornero.

**Writing – original draft:** Colton R. Martens.

**Writing – review & editing:** Colton R. Martens, Federica Accornero.

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
