## [Decision Letter · Decision Letter 0]

3 Nov 2021

Dear Dr. Accornero,

Thank you very much for submitting your manuscript "BEX1 is a critical determinant of viral myocarditis" for consideration at PLOS Pathogens. As with all papers reviewed by the journal, your manuscript was reviewed by members of the editorial board and by several independent reviewers. In light of the reviews (below this email), we would like to invite the resubmission of a significantly-revised version that takes into account the reviewers' comments.

Importantly, all reviewers highlighted the lack of the mechanistic understanding of the BEX2 effect which needs to be addressed in the revision. 

We cannot make any decision about publication until we have seen the revised manuscript and your response to the reviewers' comments. Your revised manuscript is also likely to be sent to reviewers for further evaluation.

Sincerely,

George A. Belov, PhD

Associate Editor

PLOS Pathogens

Shin-Ru Shih

Section Editor

PLOS Pathogens

Kasturi Haldar

Editor-in-Chief

PLOS Pathogens

orcid.org/0000-0001-5065-158X

Michael Malim

Editor-in-Chief

PLOS Pathogens

orcid.org/0000-0002-7699-2064

Reviewer's Responses to Questions

**Part I - Summary**

Reviewer #1: The authors of this study have demonstrated that BEX1 (Brain Expressed X-linked protein 1) is reduced in mice inoculated with coxsackievirus B3 (CVB3). BEX1 knockout (KO) mice have increased expression of CVB3 RNA in the heart as well as increased numbers of infected cardiomyocytes by day 7 implying that BEX1 expression limits the replication of these viruses in murine cardiomyocytes. In addition, the knockout mice have decreased immune cell infiltration at day 7 (normally the end of the peak of CVB3 replication in murine models of myocarditis) suggesting that BEX1 plays a role in increasing the cell mediated immune response. As might be expected, the BEX1 KO mice have decreased cardiac function and increased fibrosis at day 28 (the peak of peak of cardiac disease in the murine models of chronic myocarditis). This is interesting as it implies that the decreased inflammatory infiltration has resulted in increased damage from increased (perhaps persisting) replication of the virus due to less effective immune control of CVB3 infection. It is possible that one of the cytokines shown to be increased in the BEX1 KO mice during CVB3 infection has had effects leading to the loss of cardiac function but most of these do not persist to day 7 while the persistence of CVB3 in the heart to day 28 has been shown to correlate with susceptibility to late stage disease in the murine models of enterovirus-induced myocarditis (Chapman and Kim 2008 Curr Top Microbiol Immunol 323:275; Kandolf et al. 1999 Virus Res 62:149) as well as in human cardiomyopathy (Bouin et al. 2019 Circulation 139: 2326). It is a deficit that this study did not examine whether BEX1 is a factor in limiting the ability of the enterovirus CVB3 to persist to day 28 in the murine heart. These earlier studies in CVB3-induced myocarditis suggest that the antiviral effects of CVB3 may be more critical than the ability of BEX1 to induce infiltration of immune cells.

The earlier publication by these authors (Accornero et al. 2017) in which they demonstrated that cardiac hypertrophy and cardiac functional loss due to transverse aortic constriction could be decreased in mice by a knockout of BEX1 or increased with overexpression of BEX1 suggested that BEX1 plays a role in increasing inflammation in response to pressure stress. The present study indicates that even with knockout of BEX, a process is occurring which produces cardiac disease in these infected mice. If the infiltration of immune cells is decreased even with higher levels of CVB3 at day 7, the effect of increased fibrosis and loss of function by day 28 are likely to be mediated directly by the virus.

The authors also demonstrate that in murine embryonic fibroblast cultures, knockout of MEF increases replication of influenza and Sendai virus as well as CVB3, suggesting that the effect upon replication of CVB3 can be extended to other RNA viruses. That it can occur in the absence of an adaptive immune response is indicated by the decreased CVB3 replication in isolated cardiomyocytes from transgenic mice overexpressing BEX1. As the authors’ earlier work indicated that BEX1 may have a role in RNP complexes shuttling and stabilizing mRNAs to be translated, I would suggest BEX1 might alter the ability of the innate response to mount an antiviral effect within the cell or might affect the ability of the virus to adapt the cellular environment of the cardiomyocyte to viral translation and replication. Examination of the persistence of CVB3 in day 28 hearts with knockout or overexpression of BEX1 would be valuable in this regard. In the absence of this data, the strong indications that the effects of BEX1 upon replication of CVB3 in cardiomyocytes is more likely to underly the ability of BEX1 to decrease CVB3 induced myocarditis and heart function loss should be discussed. I disagree with the statement that "Altogether we discovered BEX1 as a novel determinant of cardiac viral infection that limits viral

replication by regulating pro-inflammatory signals and immune cell recruitment."

Reviewer #2: This paper looks at the involvement of BEX1 in the immune response in the heart after coxsackievirus B3 infection. The authors find that the immune response is dysregulated in BEX1 knockout mice after infection with the virus. While the concept is novel and interesting and potentially important, there are concerns to address.

Reviewer #3: It remains to be clarified how Bex1-ko mice with enhanced viral RNA content in the heart have impaired recruitment of immune cells. This is indeed a surprising result, particularly regarding to fibrotic injury depicted at later stages around d28. The authors should re-define their staining panels to define inflammatory immune cells in injured heart tissues, since the majority of infiltrating cells are oy myeloid origin and known to be monocytes. However, the authors failed to gate for monocytes and did not include a marker for this population. It deserves further investigation what the nature of the rise of F4/80+ cells is supposed to be after 28 days. Do these cells originate from invading monocytes? Although this might be intuitive, this interpretation is opposed by the lower abundance of inflammatory cells in Bex1-ko mice.

The manuscript lacks any information on the putative molecular function of the Bex1. The authors demonstrate the antiviral capacity by demonstrating lower viral RNA context for three infection models (in vitro, MEF, Fig.5). However, since the function of Bex1 proteins is not defined at a molecular level, the authors have to provide further information on the physiological role of this protein. In principal, their results might be attributed to altered cell survival, proliferation rate or whatever. A more detailed work-up of the infectious cycle/replication for the viruses is required to draw any conclusion on an “antiviral” function of Bex1. The interpretation particulary regarding the current pandemic SARS-CoV2 is highly speculative and not substantiated by their data.

The echocardiography data not to be shown in detail. Stroke volume, heart rate, Vol diastole, dimensions are the minimal requirements.

The authors report on Bex1-dependent alterations in cytokine profiling. This reviewer has difficulties in appreciating a consistent regulation of cytokine expression. In Fig. 2J/K, chemokine responses are higher in BEX-1ko mice, which again is not reflected in the lower immune cell abundance presented in this strain in Fig. 3.

Information on the used CVB3 virus strain is required. 129Sv/Ev mice are quite resistant to CVB3 infection. With this genetic background it is mandatory to use littermate controls for all infections.

These data are missing for the transgenic mice: echo, flow cytometry heart tissue, cytokine response, histology.

**Part II – Major Issues: Key Experiments Required for Acceptance**

Reviewer #1: (No Response)

Reviewer #2: Concerns

1) Based on the data presented it might be an overinterpretation to call bex1 a “novel anti-viral agent” (or that "that BEX1 is an antiviral regulator”) The stops short of showing a direct mechanism whereby BEX1 acts as an antiviral directly (the effect on viral infection in the KO and TG mice could be due to a number of secondary mechanisms. i.e. what if BEX1 regulates some aspect of immune cell function rather than a direct mechanism where it binds some viral component in cardiac myocytes?

2) Do bex1 tg mice have better cardiac function 28d after infection in support of the bex1 ko mice being worse off (fig. 4 d-f)? All the authors did was look at viral load 7d post infection with the tg mice.

3) How are we to know that BEX1 would be better than an anti-viral agent through immunostimulants and immunosupressors (stated in intro paragraph 3) that are currently used to treat viral myocarditis?

4) figure 2: the graphs show no error bars for these blots. If they were run on n=4, where statistics run? Furthermore, why are 3 of the cytokine array data points in the supplement?

5) Paragraph 3 of results: “Similar results were seen at day 7 for other immune cell types (Figure 3D-G Supplemental Figure 2B-C).” Fig 2B is NOT lower for bex1 ko at 7 days and 2c is barely lower. Fig 2F is lower for bex1 hearts at day 7 but is not referenced. Still, only 2 of the 18 factors tested were actually lower for bex1 hearts, a bit of an overstatement by the authors as to the implied effects.

Reviewer #3: (No Response)

**Part III – Minor Issues: Editorial and Data Presentation Modifications**

Reviewer #1: (No Response)

Reviewer #2: Minor points:

Ref 11 is for cells in culture. Is there a better ref for the cardiomyocytes being directly infected and lysed in vivo, considering that is what is trying to be referenced here?

Reviewer #3: (No Response)

PLOS authors have the option to publish the peer review history of their article (what does this mean?). If published, this will include your full peer review and any attached files.

Reviewer #1: **Yes: **Nora M. Chapman

Reviewer #2: No

Reviewer #3: No
---

## [Decision Letter · Decision Letter 1]

4 Feb 2022

Dear Dr. Accornero,

We are pleased to inform you that your manuscript 'BEX1 is a critical determinant of viral myocarditis' has been provisionally accepted for publication in PLOS Pathogens.

Best regards,

George A. Belov, PhD

Associate Editor

PLOS Pathogens

Shin-Ru Shih

Section Editor

PLOS Pathogens

Kasturi Haldar

Editor-in-Chief

PLOS Pathogens

orcid.org/0000-0001-5065-158X

Michael Malim

Editor-in-Chief

PLOS Pathogens

orcid.org/0000-0002-7699-2064

Reviewer Comments (if any, and for reference):

Reviewer's Responses to Questions

**Part I - Summary**

Reviewer #1: The authors have considerably strengthened the significance of the study with the additional experiments. It describes significant findings that should be published. I do not have further questions or suggested changes. I would like to note that another ARE-binding protein, AUF-1/hnRNP D, is cleaved by coxsackievirus B3 3C during infection by a number of enteroviruses suggesting restriction of viral RNA replication within cells. But the role of this ARE binding protein in virusreplication is not clear either.

Reviewer #2: The revisions have addressed my concerns. The work is novel and as such, represents an important contribution

**Part II – Major Issues: Key Experiments Required for Acceptance**

Reviewer #1: (No Response)

Reviewer #2: The revisions have addressed my concerns.

**Part III – Minor Issues: Editorial and Data Presentation Modifications**

Reviewer #1: (No Response)

Reviewer #2: The revisions have addressed my concerns.

PLOS authors have the option to publish the peer review history of their article (what does this mean?). If published, this will include your full peer review and any attached files.

Reviewer #1: No

Reviewer #2: No

---

## [Editor Report · Acceptance letter]

17 Feb 2022

Dear Dr. Accornero,

We are delighted to inform you that your manuscript, " BEX1 is a critical determinant of viral myocarditis ," has been formally accepted for publication in PLOS Pathogens.

Best regards,

Kasturi Haldar

Editor-in-Chief

PLOS Pathogens

orcid.org/0000-0001-5065-158X

Michael Malim

Editor-in-Chief

PLOS Pathogens

orcid.org/0000-0002-7699-2064